# Treating Mental Health and Quality of Life in Older Cancer Patients with Cognitive Behavioral Therapy: A Systematic Review and Meta-Analysis

**DOI:** 10.3390/ijerph21070881

**Published:** 2024-07-06

**Authors:** Kathryn O’Keefe, Meiyan Chen, Kevin J. Lesser, Adam S. DuVall, Alexander T. Dils

**Affiliations:** 1College of Medicine, Central Michigan University, Mt Pleasant, MI 48859, USA; okeef2kc@cmich.edu (K.O.);; 2School of Social Work, University of Michigan, Ann Arbor, MI 48109, USA; 3Section of Hematology/Oncology, University of Chicago Medicine, Chicago, IL 60637, USA

**Keywords:** cognitive behavioral therapy, older adults, geriatric cancer patients, patient-reported outcome, meta-analysis

## Abstract

Background: Cognitive behavioral therapy (CBT) has been successfully utilized in improving mental health (MH) and quality of life (QoL) in the general population, regardless of age. Cancer, which is most frequently diagnosed in older adults, is a debilitating illness that has a detrimental and long-lasting effect on patients’ MH and QoL. While numerous studies have demonstrated CBT’s efficacy, little evidence exists for its role in older cancer patients. This study, using MH and QoL metrics, evaluates the effectiveness of CBT for older adult cancer patients. Methods: Focusing on MH and QoL and an average age of over 60 years old, a final analysis was performed on 17 clinical trials with a total of 124 effect sizes, including 3073 participants receiving CBT. “Metaphor” and “Robumeta” packages in R Statistical Software (version 4.2.2) were used for analysis, which included robust variance estimation (RVE) in intercept-only meta-regression, and univariate meta-regression for moderator analysis. Results: With 17 clinical trials and 124 effect sizes, our results show that CBT moderately improves MH and QoL in cancer patients d = 0.19, 95% CI 0.0166–0.364, *p* < 0.0399. The delivery format was shown to be a strong moderator of CBT effectiveness with interpersonal technological interventions combined with pre-programmed segments having a very strong treatment effect size (d = 1.7307, 95% CI 1.5244–1.937, *p* < 0.001). Conclusions: The use of CBT in older adult cancer patients statistically improves MH and QoL, with delivery format and stages of treatment having important roles. Tech-only interpersonal interventions combined with pre-programmed CBT provide an avenue for targeting older adult cancer patients.

## 1. Introduction

Senior citizens are diagnosed with new cancers more than any other age group [1]. As our population ages and U.S. life expectancy increases, the number of geriatric patients living with cancer diagnoses will inevitably rise as well. This serves to highlight the importance of providing multifaceted care for this patient population [2]. Additionally, while new treatments are frequently being discovered and patients’ prognoses are improving, cancer continues to leave dire physiologic and psychosocial scars on patients. Emotional stressors include, but are not limited to, the looming idea of death, perceived loss of control, dependency on others, decreased self-esteem, and uncertainty about what the future may hold [3]. A patient’s psychological state is often further altered by the type of cancer as treatments may leave visible scars, producing a profound impact on body image and further exacerbating the psychosocial impact of cancer treatment [3]. This psychosocial impact has been shown to lead to short-term depression, anxiety, and post-traumatic stress disorder (PTSD) and, if left unaddressed, become long-term mental health disorders and impairments in quality of life that impact daily function in patients for years after finishing treatment [4]. Furthermore, patients currently undergoing treatment face high levels of fear of cancer progression and those who have been successfully treated face high levels of fear that the disease will reoccur [5]. This underscores the importance of addressing and treating the psychosocial impact of cancer treatment early in the disease course to provide support as patients navigate their diagnoses.

Cognitive behavioral therapy (CBT) is a tool that is commonly employed in the general population to improve quality of life metrics while also treating psychosocial disorders such as depression, anxiety, and more [6,7]. This is reinforced by the robust support for CBT effectiveness on mental health measures in the literature and bears consideration when evaluating CBT in older adults. Older adults who are diagnosed with cancer face higher rates of anxiety and depression compared to older adults in the general population [8]. CBT has been shown to be effective for mental health disorders like anxiety and depression in older adults, and while historically there is some variation in effectiveness between age groups, recent studies show no difference in effect size for older adults when compared to young and middle-aged adults [9,10]. Additionally, studies have demonstrated that older adults would rather participate in CBT than utilize medications to treat mental health disorders [11]. The effectiveness of CBT for geriatric patients combined with older adults’ positive attitudes towards using professional help make CBT a viable option for the geriatric oncology population [12,13]. Thus, using CBT in older cancer patients could be an effective avenue for treating mental health disorders and improving quality of life.

Importantly, CBT has been shown to be effective in improving the MH and QoL in cancer patients [14]. CBT can decrease the burden of diagnosis and treatment and help to ease the minds of cancer patients. When CBT is utilized properly, studies show that patients demonstrate improved insomnia symptoms, decreased anxiety and depression, improved quality of life, and ultimately, a decreased morbidity that is associated with their cancer diagnosis [15,16,17]. Addressing these psychosocial concerns of cancer patients with CBT can provide an effective and holistic approach in the treatment and survival phase of those who have been diagnosed with cancer. To date, no meta-analysis has been performed investigating CBT’s role in the treatment of older cancer patients.

Herein, we analyze the effect of CBT on MH and QoL in older cancer patients using a cutoff age of 60 years old and older. The decision to target older adults 60 and above was informed by the heterogeneity of the phrase. The World Health Organization uses the benchmark of 60 years of age to delineate older adults [18]. While some resources agree with this definition [19,20,21], others define older adults as those 75 and above [22]. To fully encapsulate those who qualify as older adults in our dataset and to accurately represent the variability inherent in health and aging, we decided to target those 60 and older [23].

Furthermore, we investigated the moderator effect of treatment modality, delivery format, number of diagnoses, and cancer treatment stage on CBT’s efficacy as it relates to MH and QoL. In the literature, Functional Scales, Symptoms Scales, Financial Well-being Scales, and Global Quality of Life are used to assess QoL. Each of these parameters primarily includes evaluations of physical, emotional, social, and cognitive functions as well as disease symptoms and treatment side effects such as insomnia, pain, or eating difficulties. MH, on the other hand, is more narrowly focused on the psychological aspects of general wellness, including depression, anxiety, psychological distress, post-traumatic stress, and general mental wellness. As such, MH and QoL outcomes were analyzed as a unit to capture a holistic view of CBT’s effect on patients’ general wellness and separately to examine different aspects of general health that could be affected by CBT.

## 2. Materials and Methods

This study followed the Methodological Expectations of Cochrane Intervention Reviews (MECIR) and reported findings in accordance with the Preferred Reporting Items for Systematic Review and Meta-Analysis (PRISMA). Our team included an interdisciplinary team of medical students, behavioral health therapist, psycho-oncologist, and research synthesis expert. We followed the protocol pre-registered at PROSPERO: CRD42020200987 but adopted an updated search date from inception to July 2023.

### 2.1. Search Procedures and Inclusion Criteria

The literature search spanned the available papers from inception to July 2023, encompassing controlled trials and including both randomized and non-randomized controlled trial studies. The search was conducted across 11 electronic databases, 4 professional websites, and a manual search of reference lists from relevant published studies. Two independent research assistants conducted the initial screening of all articles based on titles/abstracts, followed by full-text screening, utilizing the Covidence platform recommended by Cochrane for systematic reviews. Any inconsistencies in decision-making between the two independent screeners were first discussed between them to reach a consensus. If a consensus could not be reached, a senior scholar on the team cast the final vote to resolve the discrepancy.

### 2.2. Population, Intervention, and Outcome Measures

The focus was on the cancer survivor population, defined by the National Cancer Institute as individuals from the time of diagnosis throughout their remaining life span. Interventions that were solely medical or pharmaceutical in nature were not eligible for inclusion. Centering on cognitive behavioral therapy (CBT), this project considered traditionally defined CBT and several significant CBT variations. This paper adheres to Beck’s definition of CBT as a structured, present-focused psychotherapeutic approach informed by the cognitive–behavioral model, often encompassing core components such as cognitive restructuring, behavioral activation, problem-solving skills, and exposure. Consideration was given to including third-wave CBT approaches, such as acceptance and commitment-based CBT or dialectical behavioral therapy. However, this would significantly broaden the scope of the review and reduce its feasibility. The decision was made to include mindfulness-based CBT as mindfulness-based interventions are highly prevalent among cancer survivors and should be incorporated into the evidence synthesis. Given the nature of this project, the primary outcomes pertained to mental health and quality of life (QoL) among studies with the average age of participants over the age of 60 (i.e., older cancer survivors).

In our meta-analysis, mental health outcomes were defined consistently across studies to ensure uniformity. Specifically, the outcomes categorized under mental health included the following: (1) depression, (2) anxiety, (3) psychological distress, (4) post-traumatic stress, and (5) general mental wellness (e.g., as measured by the Mental Adjustment to Cancer Scale or the mental health subscale of the Short Form-12). Quality of life was consistently evaluated using the European Organisation for the Research and Treatment of Cancer Quality of Life Questionnaire and the Scale on Life Wellness, which primarily include assessments of physical, emotional, social, and cognitive functions as well as disease symptoms and treatment side effects such as insomnia, pain, or eating difficulties.

### 2.3. Data Extraction

The research team created a data extraction sheet to collect key information to facilitate data analysis. In addition to bibliographic details, the team extracted study design information (e.g., randomization method, type of comparison group, sample size), participant characteristics (e.g., mean age, percentage of female participants, percentage of non-Hispanic White participants), intervention characteristics (e.g., underlying intervention theory, core components of cognitive behavioral therapy, delivery format, etc.), and other relevant factors (e.g., whether supervision was provided or training was offered). Furthermore, the necessary statistical information was extracted to enable the calculation of effect sizes for meta-analysis.

### 2.4. Meta-Analytic Procedures

The data analysis was conducted in four phases using R software. During the initial phase, descriptive statistics summarizing the study characteristics were calculated. Subsequently, the researchers computed small sample size corrected effect size estimates for each study to determine the magnitude of the treatment effect. Since all research outcomes were continuous in nature, the between-groups standardized mean difference (SMD), also known as Hedges’ g, was calculated [24]. Following best practices, the g statistic was further adjusted with a small sample size correction to obtain an unbiased estimate of the treatment effect size, denoted as d in this study [24]. To synthesize the effect size estimates across the included studies, meta-regression with robust variance estimation (RVE) was employed [25,26]. Meta-regression with RVE was chosen because it effectively utilizes an intercept-only model to provide an overall average of treatment effect sizes across studies. This analytical approach not only handles dependent effect sizes (i.e., when more than one effect size estimate is reported within a single study and all reported effect size estimates are included) but also produces robust statistical inference regardless of the variance modeling strategy employed, whether fixed- or random-effects modeling [25]. Finally, the researchers planned to conduct subgroup analyses and univariate moderator analyses based on outcome categories. The “metaphor” and “robumeta” packages of R Statistical Software (version 4.2.2) were utilized for all data analyses.

## 3. Results

### 3.1. Study Characteristics

The final analysis of all the studies was restricted to those investigating MH and QoL and to studies whose participants had an average age of 60 years and older. The analysis encompassed 17 trials, comprising 124 effect size estimates and a total of 3073 participants receiving cognitive behavioral therapy (CBT) interventions. The range of the average age amongst each of the 17 individual studies was from the lowest average age of 60.04 to the highest average age of 76 years old. Among the 17 trials that reported patient gender, 48.29% of the participants were female (*n* = 1484). With the exception of a single study, all trials were randomized controlled trials (RCTs), while one trial was a non-randomized controlled trial. In the present review, of the eleven studies that reported intervention frequency, the majority implemented a weekly intervention regimen. More than 40% of the studies (7 out of 17) employed group-based interventions as the primary method, while seven studies utilized individual-based interventions as their strategy, and two studies adopted family-based interventions as their treatment modality (Figure 1).

### 3.2. Publication Bias

To assess the potential presence of publication bias, both a funnel plot and the Vevea and Woods (2005) sensitivity analysis employing a weight-function model were utilized [27]. The funnel plot (Figure 2), which visually exhibited an absence of data points with lower standard errors, suggested the possible existence of publication bias in the current study. However, the application of the Vevea and Woods weight model did not provide evidence to support the presence of publication bias. The observed overall treatment effect size was not found to be statistically different from the theoretical overall treatment effect size, assuming a symmetric funnel plot without publication bias.

### 3.3. Risk of Bias

In this study, the risk of bias was evaluated using the Revised Cochrane Risk-of-Bias tool 2nd version (Appendix A) and the Non-Randomised Studies of the Effects of Interventions (ROBINS-I) (Appendix A) for both randomized controlled trials (16/17) and non-randomized controlled trials (1/17). In general, the studies included in this review identified a low risk of bias in appropriate reporting in the measurement of the outcomes (17/17), concern in the selection of the reported results (17/17), reporting in the randomization process (17/17), and deviations from the intended interventions exited among studies (17/17). However, three randomized controlled trials reported a moderate risk of bias addressing missing outcome data (1/17), and one non-randomized controlled trial indicated a moderate risk of bias in the selection of participants into the study. Overall, the studies incorporated in this review exhibited a relatively low risk of bias.

### 3.4. Meta-Analytic Results

Across the 17 studies (containing 214 effect sizes) included in this meta-analysis, the pooled treatment effect size was d = 0.19, 95% CI [0.0166, 0.364], and *p* = 0.0399, indicating a statistically significant treatment effect of CBT interventions for older adults’ mental health outcomes and quality of life outcomes. In comparison to the control condition, participants who received interventions were, on average, 0.19 standard deviations better (improved) (Table 1). For subgroup analyses based on cancer stage, CBT interventions for ongoing curative treatment, g = 0.173, 95% CI [0.0117, 0.335], and *p* = 0.0388, reported overall statistically significant treatment effects, respectively (Table 1). Correspondingly, the rest of the subgroups were not statistically significant (Table 1). For subgroup analyses based on outcome categories, an overall significant treatment effect was observed only among the mental health outcome, g = 0.217, 95% CI [0.00662, 0.426], and *p* = 0.044 (Table 1).

### 3.5. Moderator Analysis

The moderators examined included the following: (1) treatment modality (individual based; family based; individual based combined with small group based; small group based); (2) delivery format (in-person; mixed in-person and tech; tech-only interpersonal and pre-programmed; tech-only interpersonal); (3) number of diagnoses (multiple versus single); (4) cancer stage (mixed; not described; ongoing curative treatment; post-treatment survivorship); (5) outcome categories (quality of life versus mental health) (Table 2). Univariate meta-regression analysis using a single predictor discovered that the treatment modality, number of diagnoses, cancer stage, and outcome categories were not significant moderators (Table 2). When compared to the in-person delivery format, tech-only interpersonal and pre-programmed (b = 1.7307, *p* = 0.272), self-wellness outcomes (b = −0.349, *p* = 0.396) and general health outcomes (b = −0.492, *p* <0.001) had significantly different treatment effects (Table 2).

## 4. Discussion

CBT’s strong effect on mental health and quality of life metrics has been well demonstrated over the years, yet discussion continues as to the efficacy of CBT for older cancer patients. The current literature clearly demonstrates that CBT is still largely effective in older adults across multiple domains [28,29,30]. Given the high incidence and prevalence of cancer in older adults, we investigated the efficacy of CBT in this geriatric oncology population. Our results demonstrate that CBT has a combined effectiveness on MH and QoL with an overall treatment effect size of 0.19, highlighting the utility of CBT interventions in older cancer patients.

Upon moderator analysis, the delivery format setting proved to have a strong, clinically significant effect on oncology patients over the age of 60, targeting a potential mode of intervention for this geriatric population. Strikingly, tech-only interpersonal combined with pre-programmed CBT provides a significant and large treatment effect size of 1.7 as compared to in-person treatment. Conversely, neither treatment modality, number of diagnoses, cancer staging, nor outcome metrics proved to be significant as individual moderators of CBT efficacy. This bears considerable weight in the discussion of strategies to improve CBT in older oncology patients.

Research in this area remains limited but is largely in agreement with these findings. Choi et al. found that problem-solving therapy provided via telehealth had longer-lasting effects for older adults than that provided in person [31]. Xiang et al. suggest that internet-based CBT may be even more effective for older adults despite the paucity of data for this population [32]. The severe lack of therapists, the fear of potential stigmatization, geographic isolation, long wait times, and high costs can all lead to remote CBT being more effective in treatment [33]. In fact, studies have shown that technology can successfully improve the QoL of older patients en masse [34]. Additionally, in-person CBT carries multiple obstacles for cancer patients to receive mental health treatment. This is highlighted by the rigors of cancer treatment imposed on patients, such as restricted mobility, chronic pain, and lack of access to reliable transportation. Therefore, making remote CBT an accessible option for patients could provide an avenue for overcoming these barriers to mental health treatment. In fact, Greer et al. developed a mobile app to provide CBT to oncology patients and found it to be effective in patients with incurable cancers [14]. Such interventions show that the field is already moving in this direction and provide a framework for future interventions. Furthermore, Greer et al. called for the integration of live therapy sessions alongside pre-programmed therapies, and others have postulated that the interpersonal and pre-programmed method could reinforce adherence and provide needed guidance for unaided portions of CBT, all of which further corroborates our findings [14,32]. Combined, we propose that technology-based CBT with both pre-programmed and interpersonal methods can effectively target older oncology patients.

We next sought to identify the effects of CBT on MH and QoL separately using subgroup analysis. We found that the effect of CBT on MH was statistically significant (*p*-value of 0.044), whereas its impact on QoL was not (*p*-value 0.29), suggesting that CBT is effective in older cancer patients for MH, whereas QoL may be less malleable in older oncology patients.

Many factors could contribute to these findings. QoL is a multifaceted measurement that takes many parameters into consideration, including certain physical aspects that may not be alleviated by CBT (see aforementioned methods on QoL parameters). Thus, we propose that the rigors of cancer treatment could leave lasting and profound impacts that CBT is not equipped to overcome. This is supported by the literature evaluating the loss of functional status, demonstrated as Activities of Daily Living (ADLs) and Instrumental ADLs (IADLs), in older cancer patients with colorectal cancer and was found to correlate with QoL deterioration [35]. Studies by Rønning et al. and Meert et al. corroborate these findings by demonstrating that ADLs and IADLs in older colorectal cancer patients were significantly decreased during treatment [36,37]. These studies indicate that there is limited capacity for QoL improvement in older oncology patients as cancer and treatments themselves cause lasting harm to QoL metrics. To mitigate these harms, the oncogeriatric assessment tool, which aims to reduce the toxic effects of cancer treatment in older oncology patients, is gaining traction in clinical practice.

By utilizing the oncogeriatric assessment, providers can strategize therapy to avoid the harmful side effects of treatment by identifying functional and psychosocial deficits prior to treatment initiation [38]. In fact, recent developments of the oncogeriatric assessment have been shown to reduce toxic effects in older cancer patients 70 and above [39]. However, this tool is still under development and 2023 guidelines mark the beginning of efforts to standardize the assessment, citing inconsistent uptake in clinical practice [40]. Thus, the use of the oncogeriatric assessment was limited in our dataset, unable to be quantified, and not considered. Additionally, its current use is limited to cancer patients 65 and older, a sparsely studied population regarding CBT’s efficacy. It is, however, a valuable tool that could reduce the impact of cancer treatment on patients, creating a world in which CBT may have an equal effect on QoL and MH when controlled for pre-existing functional and psychosocial deficits. Regardless, given the limited data on QoL in older cancer patients, further studies are required to understand the full impact of CBT in this population.

Subgroup analysis of CBT’s effectiveness on combined MH and QoL in older oncology patients in different stages of treatment revealed that CBT is effective only in patients currently undergoing treatment and notably not effective after treatment was concluded. This highlights the extensive impact that cancer treatment has on a patient’s psychological well-being. As discussed previously, older cancer patients suffer considerable effects from cancer treatments and thus likely need additional assistance for the duration of their treatment, but not necessarily after the treatment has concluded. This is augmented by the literature demonstrating that older adults have more emotional resilience which is built through years of adversity and they can fall back on this once the emotional strain of cancer treatment has concluded [11]. Thus, older cancer patients benefit from CBT therapy the most when these coping mechanisms that are built through resilience fail due to the psychological toll of cancer treatment itself [41]. Upon our investigation using univariate moderator analysis, a comparison between the treatment modalities revealed no statistical difference. Given the severe lack of research on the older cancer patient population, the absence of significance serves to reinforce the need for more data on the geriatric oncology population to investigate the appropriate time frame of CBT intervention adequately. Our data identify a need to optimize CBT treatment for older oncology patients for the duration of their treatment and likely the continuation of CBT treatment for some time thereafter.

Currently, there is an abundance of research aimed at delineating CBT treatment effectiveness in oncology patients, which is highlighted by the vast number of records and treatment effect outcomes in our data search. However, our analysis investigating MH and QoL outcomes in studies with an average age of 60 years old and older was sparse. While this highlights a limitation of our study, it also demonstrates how underrepresented this patient population is given that the median age of patients receiving cancer diagnoses hovers around 66 years old [1]. The proportion of Americans older than 65 is expected to increase to almost a quarter of the U.S. population by 2050, and as cancer mortality decreases by roughly 2% each year, the prevalence of older adults who have had cancer diagnoses will increase as well [42,43]. This increasing prevalence conveys the importance of having robust treatment options that target the mental health aspects of disease and wellness, especially for an age group that makes up half of new cancer diagnoses. It is imperative, then, that we have more research into geriatric cancer patients and methods to improve psychosocial wellness during cancer treatment.

## 5. Conclusions

It has been well established that the mental toll a cancer diagnosis and subsequent treatment can have on patients is devastating. Our data show that the integration of CBT into oncology treatment can greatly improve mental health in older cancer patients. Specifically, our findings indicate that CBT is an effective intervention for oncology patients 60 and older, with a strong treatment effect size for mental health and a very strong moderator effect for tech-only interpersonal and pre-programmed interventions combined. Furthermore, we demonstrated that using CBT is effective during treatment in older cancer patients. We believe this provides an avenue for CBT in a population that has been largely left out of the intervention. Further research is needed regarding the psychosocial impact of cancer treatment in geriatric oncology patients as it provides clinical relevance to the therapeutic advantages of technological-based CBT interventions in this older cancer patient population, as they make up an ever-increasing proportion of our population and are a large number of cancer diagnoses each year.

## Figures and Tables

**Figure 1 ijerph-21-00881-f001:**
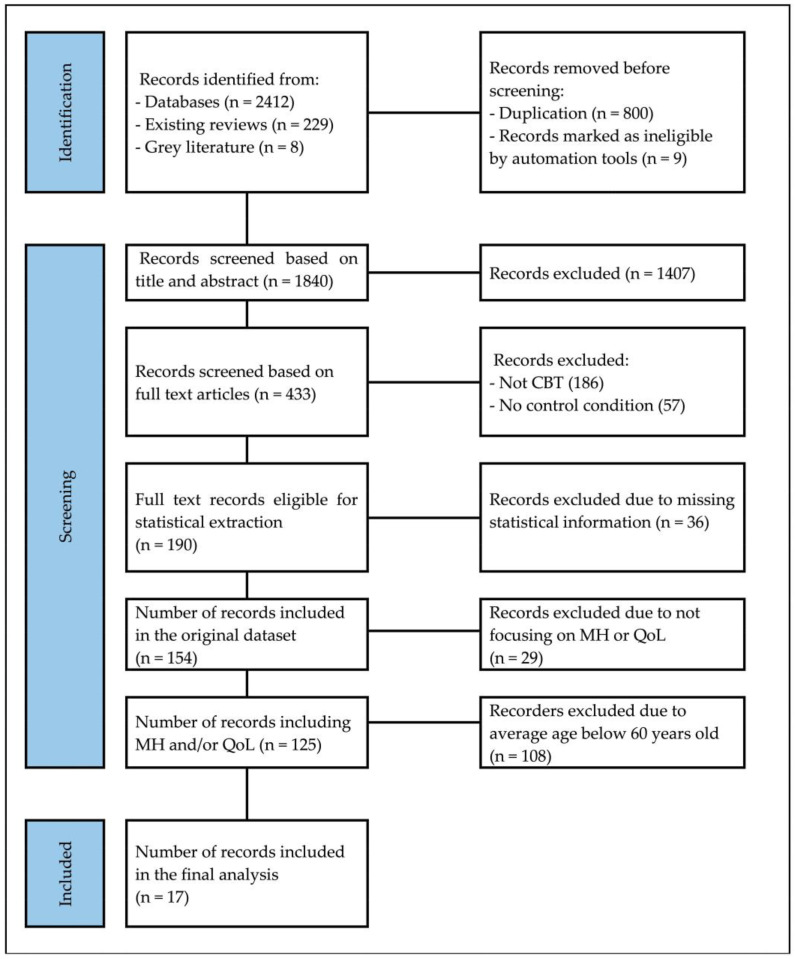
PRISMA 2020 flow diagram for literature search.

**Figure 2 ijerph-21-00881-f002:**
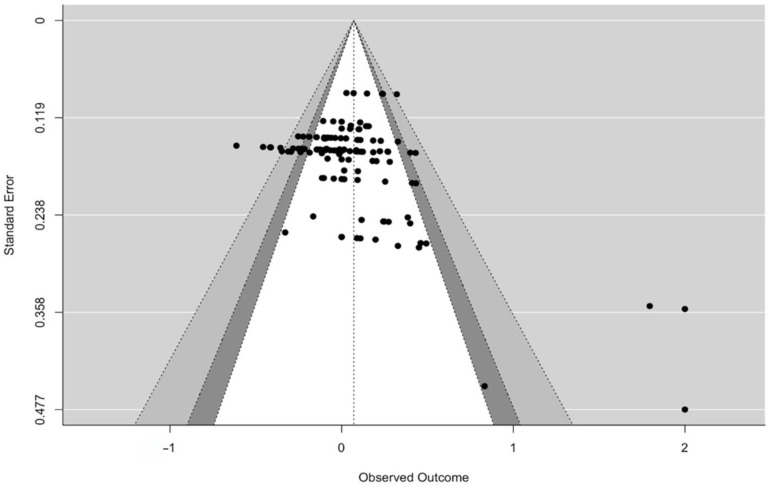
Funnel plot for publication bias. Colors correspond to confidence interval region; white is 90, dark gray is 95, and light gray is 99.

**Table 1 ijerph-21-00881-t001:** Overall treatment effect and subgroup analysis.

Estimate		N/K	df	95% CI	*p* Value
Overall	0.19	17/124	15	0.0166–0.364	0.0399 *
Subgroup analysis with treatment modality					
Individual Based	0.27	10/44	8.45	−0.0621–0.602	0.0983
Small group based	0.038	6/65	4.45	−0.129–0.205	0.573
Individual Based Combined with Small Group Based	Due to the small sample size (1/4), we are not able to analyze this subgroup.				
Family Based	Due to the small sample size (1/9), we are not able to analyze this subgroup.				
Subgroup analysis with the delivery format					
In-person Therapy	0.17	8/61	6.24	−0.0336–0.373	0.0876
Mixed In-person and Tech	0.103	4/22	2.63	−0.131–0.337	0.239
Tech-only Interpersonal	0.0384	5/39	3.73	−0.205–0.282	0.678
Technology Interpersonal and Pre-programmed	Due to the small sample size (1/2), we are not able to analyze this subgroup.				
Subgroup analysis with the number of diagnoses					
Multiple	0.238	5/24	3.64	−0.297–0.773	0.275
Single	0.182	12/100	10.5	−0.0308–0.394	0.0862
Subgroup analysis with cancer treatment stage					
Ongoing Curative Treatment	0.173	9/59	7.21	0.0117–0.335	0.0388 *
Post-treatment Survivorship	0.679	3/18	1.99	−1.71–3.07	0.344
Mixed	−0.0369	3/39	1.9	−0.721–0.648	0.831
Not described	Due to the small sample size (2/8), we are not able to analyze this subgroup.				
Subgroup analysis with outcome					
Mental Health	0.217	16/84	14.2	0.00662–0.426	0.044 *
Quality of Life	0.686	8/40	5.81	−0.0766–0.214	0.29

K = number of studies; N =  number of effect size estimates; df, degrees of freedom. If dfs < 4, a lower *p*-value (*p* < 0.01) should be used as a threshold for statistical significance; CI = Confidence Interval; * *p* < 0.05.

**Table 2 ijerph-21-00881-t002:** Univariate moderator analyses.

Estimate		N/K	df	95% CI	*p* Value
Treatment modality (ref: Individual based)	0.282	16/122	8.01	−0.0591–0.623	0.093
Family based	−0.177	16/122	8.01	−0.5184–0.164	0.265
Individual based combined with small group based	−0.157	16/122	8.01	−0.4982–0.184	0.319
Small group based	−0.227	16/122	9.55	−0.6058–0.152	0.211
Delivery format (ref: In-person)	0.1669	17/124	6.22	−0.0394–0.373	0.09556
Mixed in-person and tech	−0.0145	17/124	5.11	−0.02887–0.260	0.89769
Tech-only interpersonal and pre-programmed	1.7307	17/124	6.22	1.5244–1.937	<0.001 ***
Tech-only interpersonal	−0.1290	17/124	8.97	−0.4015–0.143	0.31168
Number of Diagnoses (ref: Multiple)	0.2196	17/214	3.54	−0.292–0.731	0.286
Single	−0.0345	17/214	6.44	−0.518–0.449	0.869
Cancer stage (ref: Mixed)	−0.0252	17/214	1.94	−0.736–0.686	0.890
Not described	0.1048	17/214	2.31	−0.510–0.719	0.577
Ongoing curative treatment	0.2235	17/214	3.37	−0.316–0.763	0.294
Post-treatment survivorship	0.5693	17/214	3.73	−0.800–1.938	0.305
Outcome (ref: Quality of life)	0.103	17/124	4.55	−0.0826–0.289	0.206
Mental Health	0.112	17/124	6.66	−0.1353–0.360	0.316

K = number of studies; N =  number of effect size estimates; df, degrees of freedom. If dfs < 4, a lower *p*-value (*p* < 0.01) should be used as a threshold for statistical significance; CI = Confidence Interval; *** *p* < 0.001.

## Data Availability

Raw data can be made available upon reasonable request from the corresponding author.

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
