# Peer review of "Treating Mental Health and Quality of Life in Older Cancer Patients with Cognitive Behavioral Therapy: A Systematic Review and Meta-Analysis"

_ijerph, 2024, doi:10.3390/ijerph21070881_

Round 1
Reviewer 1 Report
Comments and Suggestions for Authors
I first thank the authors for the topic of interest. However, there are, in my opinion, issues in the article that need to be clarified or corrected.
Major comments
The authors selected clinical trials where the mean age was 60 years or older. Primarily, according to the vast majority of authors in the literature, geriatric patients are considered those who are older than 65 years. Why did the authors choose this cut-off? Furthermore, by selecting clinical trials on the basis of the mean age of 60 years, can younger patients also be present within individual trials? How was this issue resolved? What percentage of patients older than 60 years are present in these clinical trials? This selection issue risks undermining the generalization of the data for geriatric patients only.
Mental health and quality of life were chosen as outcomes. It does not appear clear how these outcomes were quantified in the selected clinical trials. Are the ways of measuring these outcomes uniform across trials? If not, how could this issue be resolved by the authors? Why were they put together as a single outcome? These points need clarification in the text.
Minor comments
I think the abstract should be reworded to make the message of the article clearer and more consistent.
The authors often use the word elderly, which is not polite. Please use a more appropriate term.
References to oncogeriatric assessment in the oncology setting and how these interventions can fit into the matter of the article were not included.
There are references in the discussion to the difficulty of improving quality of life for older adults due to the increased disability associated with cancer treatments. Even if this issue is real, the phrase, according to me, is not so clear and could lead to the idea that it is not possible to improve quality of life in older adults.
The discussion mentions how older adult patients are often excluded or not well represented in clinical trials. This sentence, although real and correct, is expressed in a very unclear way.
Comments on the Quality of English Language
There are some minor grammatical errors that have to be fixed.
Author Response
Reviewer #1:
I first thank the authors for the topic of interest. However, there are, in my opinion, issues in the article that need to be clarified or corrected.
Dear reviewer,
Thank you very much for taking the time to read our manuscript and provide feedback on points of correction and clarification. Below you will find our replies to your comments and we hope that we have successfully answered the concerns that you highlighted. Attached to this reply are the changes (see highlighting) that we have made to the manuscript as a direct response to your concerns.
Major Comments:
The authors selected clinical trials where the mean age was 60 years or older. Primarily, according to the vast majority of authors in the literature, geriatric patients are considered those who are older than 65 years.
- Why did the authors choose this cut-off?
- The selected age of 60 and older was based on the idea that many organizations and much of the literature considers older adults as 60 and above. Much of the explanation as to why this is the cut-off is further elaborated upon and supported in the fourth paragraph of the introduction.
- Furthermore, by selecting clinical trials on the basis of the mean age of 60 years, can younger patients also be present within individual trials? How was this issue resolved? What percentage of patients older than 60 years are present in these clinical trials? This selection issue risks undermining the generalization of the data for geriatric patients only.
- Thank you for your feedback. Regarding your concern about the selection of clinical trials based on the mean age of 60 years, I would like to explain that our initial inclusion criteria specifically targeted studies with populations aged 60 years and older. Consequently, studies focusing on younger patients were excluded during the preliminary screening process so the issue of including younger patients within individual trials does not affect our analysis. The screening process ensures that the data is specifically targeted toward analyzing mental health and quality of life issues in the geriatric population.
Mental health and quality of life were chosen as outcomes. It does not appear clear how these outcomes were quantified in the selected clinical trials.
- Are the ways of measuring these outcomes uniform across trials? If not, how could this issue be resolved by the authors?
- Thank you for your comments. Regarding the quantification of mental health and quality of life outcomes in the selected clinical trials, I have now included a detailed explanation in the manuscript which can be found in section 2.2.
- Why were they put together as a single outcome? These points need clarification in the text.
- We wanted a holistic view of CBT’s effects in older cancer patients so we included quality of life which considers a wide array of aspects that contribute to an individual’s well-being including but not limited to physical, emotion and financial health. We mistakenly did not fully explain what quality of life entails in this manuscript but we hope that our edits provide clarity. Changes within the manuscript can be found at the end of the introduction and within the discussion.
Minor comments:
- I think the abstract should be reworded to make the message of the article clearer and more consistent.
- The abstract has been rewritten to provide more clarity. Please see the newly written abstract.
- The authors often use the word elderly, which is not polite. Please use a more appropriate term.
- We would like to apologize for using this term as we were not aware it was impolite. As such, we have changed the wording throughout the document to be more respectful of the patient population so any reference to the patient population is as “older adults” or “older oncology patients” or “older cancer patients”
- References to oncogeriatric assessment in the oncology setting and how these interventions can fit into the matter of the article were not included.
- Thank you for pointing this out to us. Prior to writing this manuscript, we were not aware of the oncogeriatric assessment in the oncology setting. Upon reading further into this, we realized this is an extremely important assessment that could be utilized in treating older adults with cancer so their psychosocial health is taken into consideration. Since this tool is newer and the guidelines were not established until 2023, all of the trials we analyzed did not take this tool into consideration so it has minimal effect on our analysis. However, we agree that this is an important tool to comment on and have made note of it in the manuscript (see section 3 Discussion, paragraph 5 and 6) and highlighted that it should be taken into consideration when assessing the effects of CBT in older adults.
- There are references in the discussion to the difficulty of improving quality of life for older adults due to the increased disability associated with cancer treatments. Even if this issue is real, the phrase, according to me, is not so clear and could lead to the idea that it is not possible to improve quality of life in older adults.
- We reworded some of our language in this section of the discussion to provide clarity. In short, our message is that while CBT is proven to be effective in improving quality of life metrics in the general population including older adults, our data suggests that it may not be as effective in improving quality of life in older adults with cancer. There are many reasons why this could be the case but we hypothesize that this is likely due to the adverse effects of cancer treatment on this patient population. Please see section 3 Discussion, paragraph 5 and 6 for how we explained this matter.
- The discussion mentions how older adult patients are often excluded or not well represented in clinical trials. This sentence, although real and correct, is expressed in a very unclear way.
- his sentence has been reworded to provide more clarity.
- There are some minor grammatical errors that have be to fixed.
- Thank you for noticing this. We have made changes throughout the document and hope we have captured all of the grammatical errors.
Reviewer 2 Report
Comments and Suggestions for Authors
The study is well-written and presents a rationale favorable to the objective. Its methods are coherent and consistent with what is expected in a systematic review and meta-analysis.
I have only two recommendations:
1 - Insert into a table or within the text the PICOS framework used to guide the screening.
2 - In addition to the tables, I recommend including a classic meta-analysis graph with the individual results of each study.
Author Response
Reviewer #2:
The study is well-written and presents a rationale favorable to the objective. Its methods are coherent and consistent with what is expected in a systematic review and meta-analysis.
Dear reviewer,
We are delighted to hear that you found our manuscript to be well-written and favorable to the objective! We also thank you very much for taking the time to read our manuscript and provide feedback. Below you will find our replies to your comments and we hope that we have successfully answered the concerns that you highlighted. Attached to this reply are the changes (see highlighting) that we have made to the manuscript as a direct response to your concerns.
I have only two recommendations:
- Insert into a table or within the text the PICOS framework used to guide the screening.
- Within the text, we expanded on our explanation of what our study contained and provided more clarity as to the intervention (CBT), comparison (no CBT) and outcomes (MH and QoL). We hope that by expanding on our language in the manuscript, the PICOS framework is appropriately outlined.
- In addition to the tables, I recommend including a classic meta-analysis graph with the individual results of each study.
- We appreciate your thought on this matter. However, given the scope of the manuscript and the extensive number of clinical trials that were analyzed, this would be a significant task and is currently not feasible.
Reviewer 3 Report
Comments and Suggestions for Authors
Thank you for the opportunity to review this manuscript. The study offers valuable insights into evaluating CBT's efficacy in older cancer patient populations where evidence is scarce and adds significant value to the existing literature. I only have some minor concerns and by addressing them, the quality and clarity of the manuscript will be strengthened:
Methods:
1. Please provide the rationale for using an average age of 60 as the cut-off age.
2. Please use RoB2 to conduct a risk of bias assessment and report the results.
Table: Please indicate where each table was inserted in the text.
Author Response
Reviewer #3
Thank you for the opportunity to review this manuscript. The study offers valuable insights into evaluating CBT's efficacy in older cancer patient populations where evidence is scarce and adds significant value to the existing literature.
Dear reviewer,
We are excited to hear that you found our manuscript valuable and provide contribution to this field! We also thank you very much for taking the time to read our manuscript and provide feedback. Below you will find our replies to your comments and we hope that we have successfully answered the concerns that you highlighted. Attached to this reply are the changes (see highlighting) that we have made to the manuscript as a direct response to your concerns.
I only have some minor concerns and by addressing them, the quality and clarity of the manuscript will be strengthened:
Methods:
- Please provide the rationale for using an average age of 60 as the cut-off age.
- The selected age of 60 and older was based on the idea that many organizations and much of the literature considers older adults as 60 and above. Much of the explanation as to why this is the cut-off is further elaborated upon and supported in the fourth paragraph of the introduction.
- Please use RoB2 to conduct a risk of bias assessment and report the results.
- This has been completed and we are including the RoB2 table as a supplemental tables which can be located at the end of the edited manuscript (see attached)
- Table: Please indicate where each table was inserted in the text.
- We have included where each table was inserted into the text.
Round 2
Reviewer 1 Report
Comments and Suggestions for Authors
I believe the authors have made a significant overall improvement in the article, clarifying several points and enhancing the text as a whole.
However, I think the study design still needs further clarification. Specifically, regarding the age of the patients enrolled in the selected clinical trials, the text states in the results that the included studies had an average age ranging from 60.04 to 76.00 years. Therefore, at least some of these studies must have included patients younger than 60 years. Additionally, the PRISMA 2020 Flow Diagram for Literature Search indicates that the criteria for arriving at the final 17 studies was an average age below 60 years. This makes it unclear how patients under 60 years old, who I believe are present in the selected studies according to the exclusion criteria, were excluded from the analysis. Did you do a subanalysis? If this is the case, no description of it is present in the text.
As a minor comment, I would also appreciate consistency in the number of digits used after the comma throughout the text and in tables.
Author Response
Reviewer #1 (2nd comments):
I believe the authors have made a significant overall improvement in the article, clarifying several points and enhancing the text as a whole.
Dear reviewer,
We are happy to hear that you have found significant improvement in the article and clarity is been provided! We would also like to thank you for providing your feedback on areas that could be improved. Below you will find our replies to your comments and we hope that we have successfully answered the concerns that you highlighted. Attached to this reply are the changes (see highlighting) that we have made to the manuscript as a direct response to your concerns.
However, I think the study design still needs further clarification.
- Specifically, regarding the age of the patients enrolled in the selected clinical trials, the text states in the results that the included studies had an average age ranging from 60.04 to 76.00 years. Therefore, at least some of these studies must have included patients younger than 60 years.
- While the ranges of participants ages in each study varies, we analyzed studies that reported an average age of 60 years old and older. While some articles may report all of the ages of the participants, we do not have all of the raw data from each article to reanalyze the efficacy of CBT on MH and QoL in cancer patients only 60-years-old and older. As such, we believe that our meta-analysis of the trials reported average age demonstrates how the therapeutic tool of interest we are investigating (ie CBT) impacts different outcomes (ie MH and QoL) in the previously defined older adult patient population.
- Additionally, the PRISMA 2020 Flow Diagram for Literature Search indicates that the criteria for arriving at the final 17 studies was an average age below 60 years. This makes it unclear how patients under 60 years old, who I believe are present in the selected studies according to the exclusion criteria, were excluded from the analysis. Did you do a subanalysis? If this is the case, no description of it is present in the text.
- Thank you for noticing this. Within the original Flow Diagram, our intention was to demonstrate that after identifying the articles that use CBT to treat MH and QoL, we separated the articles based on the articles reported average age of participants. From there, we took articles that reported their participants age as 60 years old or older and analyzed the effects of CBT on treating MH and QoL. We have reworded some of the Flow Diagram to provide more clarity.
- As a minor comment, I would also appreciate consistency in the number of digits used after the comma throughout the text and in tables.
- With regard to the consistency in digits, are you asking after a decimal point? We can easily change the text to an equal number of digits within the provided text and tables after each decimal point. Please let us know if this is what you meant and we will make the change!